# O-Researcher: An Open Ended Deep Research Model via Multi-Agent Distillation and Agentic RL

## Abstract

The performance gap between closed-source and open-source large language models (LLMs) is largely attributed to disparities in access to high-quality training data. To bridge this gap, we introduce a novel framework for the automated synthesis of sophisticated, research-grade instructional data. Our approach centers on a multi-agent workflow where collaborative AI agents simulate complex tool-integrated reasoning to generate diverse and high-fidelity data end-to-end. Leveraging this synthesized data, we develop a two-stage training strategy that integrates supervised fine-tuning with a novel reinforcement learning method, designed to maximize model alignment and capability. Extensive experiments demonstrate that our framework empowers open-source models across multiple scales, enabling them to achieve new state-of-the-art performance on the major deep research benchmark. This work provides a scalable and effective pathway for advancing open-source LLMs without relying on proprietary data or models.

## 1 Introduction

The rapid evolution of Large Language Models (LLMs) has become a cornerstone of modern artificial intelligence, driving breakthroughs in tasks ranging from natural language understanding to complex reasoning. A persistent challenge within this field, however, is the performance disparity between powerful, closed-source models (e.g., GPT-4) and their open-source counterparts. This gap is often attributed to the vast amounts of proprietary, high-quality training data and immense computational resources available to their developers. Consequently, the research community faces a significant bottleneck: how to effectively empower open-source models to achieve state-of-the-art (SOTA) performance without relying on such exclusive advantages.

A critical pathway to overcoming this bottleneck is the generation of high-quality, diverse, and scalable instructional data for supervised fine-tuning (SFT) and reinforcement learning (RL). While existing methods often rely on human annotation or distillation from larger teacher models, these approaches are either prohibitively expensive, limited in scale, or risk inheriting the limitations of the teacher. There is a pressing need for an automated, end-to-end framework that can synthesize vast and sophisticated research-grade data to fuel the next generation of open-source LLMs.

In this paper, we propose a novel framework that addresses this fundamental challenge through a multi-agent driven synthetic data generation workflow. Our core insight is that a collaborative ecosystem of specialized AI agents can autonomously simulate complex human reasoning processes to create high-fidelity, multi-turn instruction-response pairs. This synthetic data serves as the foundation for a robust two-stage training strategy, which integrates a novel reinforcement learning method designed to align model outputs with precise quality and correctness metrics.

Our contributions are summarized as follows:

- We raise a novel multi-agent driven workflow to automatically synthesize end-to-end deep research data generation. This system leverages a structured collaboration between multiple LLM agents to decompose, debate, and verify complex tasks, resulting in a scalable pipeline for producing premium training corpora.
- Based on the synthesized data from our proposed workflow, we design a two-stage training strategy that incorporates a novel reinforcement learning method. This strategy first employs supervised fine-tuning on the synthetic data, followed by a reinforcement learning phase that further refines the model's performance.
- We demonstrate that our method empowers open-source models in multiple sizes to achieve new SOTA performance. Through extensive experiments on the major deep research benchmark, we show that models trained with our framework not only significantly close the gap to closed-source models but also establish new state-of-the-art results for models of comparable size in the open-source domain.

## 2    RELATED WORK

Large Reasoning Models (LRMs) Achiam et al. (2023); Guo et al. (2025); Yang et al. (2025a) have achieved impressive results even in challenging domains like math and code. Web agents have also become a significant research direction, with rapid development driven by systems such as WebSailor Li et al. (2025b), WebSailor-V2 Li et al. (2025a), WebWeaver Li et al. (2025f), WebDancer Wu et al. (2025a), and Web-Thinker Li et al. (2025d), each proposing more powerful strategies for web-scale reasoning and information-seeking. Nevertheless, their capabilities remain fundamentally constrained by their internal closed-world knowledge boundaries (Zhang et al., 2025b). To mitigate these limitations, agentic systems Yang et al. (2023); Pandya & Holia (2023) have been introduced to augment LRMs by enabling interaction with external APIs, search engines, and computational tools, etc. Wu et al. (2025b). In December 2024, Google introduced its initial research-oriented implementation, Gemini Deep Research Google Team (2025), focusing on the agentic system with multi-step reasoning and knowledge integration. Building on these advancements, the task of Deep Research Report Generation Zheng et al. (2025; 2024); Yang et al. (2025b) emerged as a specialized benchmark designed to systematically exploit agentic capabilities for producing evidence-based analytical reports that address complex research questions. Such research tasks are inherently challenging, as they typically require capabilities of executing long-horizon planning, performing multi-hop information retrieval, and iteratively invoking external tools, ultimately synthesizing heterogeneous, web-sourced evidence into comprehensive, faithful, and well-structured analyses Zhang et al. (2025b); Xu & Peng (2025); Huang et al. (2025). In addition, DeepResearchGym Coelho et al. (2025) provides an open, reproducible sandbox for evaluating these abilities across diverse long-horizon scenarios.

Despite these challenges, the field is advancing rapidly, driven by proprietary DRAs such as OpenAI Deep Research OpenAI (2025), Tongyi Deep Research Tongyi DeepResearch Team (2025); Tongyi Lab (2025), Grok DeeperSearch xAI Team (2025) , and AutoGLM-Research Zhipu AI (2025), alongside open-source efforts including Jina-AI node-DeepResearch Jina AI (2025), Agentic Reasoning Wu et al. (2025b), Deep-Researcher Zheng et al. (2025), OpenResearcher Zheng et al. (2024), and WebLeaper Tao et al. (2025). Collectively, these systems are centered around four interconnected aspects: planning, query formulation, knowledge discovery, and report generation (knowledge synthesis) Zhang et al. (2025b); Xu & Peng (2025). Mainstream DRAs Yang et al. (2025b); Wu et al. (2025b) commonly employ a monolithic architecture, where a single LRM conducts planning, generates queries, and invokes external tools for knowledge discovery. Such tools include API-based web search and coding environments Schmidgall et al. (2025); Wu et al. (2025b), enabling LRMs to overcome internal knowledge limitations and perform interactive reasoning. To ensure robust planning across extended reasoning chains with frequent tool use, Agentic Reasoning Wu et al. (2025b) incorporates a Mind-Map agent that dynamically constructs a knowledge graph. In terms of knowledge synthesis, Agent Laboratory Schmidgall et al. (2025) employs structure-controlled, planning-based generation, while Multimodal DeepResearcher Yang et al. (2025b) extends beyond text-only outputs

to produce interleaved multimodal reports combining text and charts. Parallel to these developments, search-enhanced reasoning methods such as Search-R1 Jin et al. (2025) and Search-o1 Li et al. (2025e) propose reinforcement learning pipelines that explicitly couple search behavior with reasoning trajectories to improve factual grounding and retrieval efficiency. Most prior works remain prompt-based search agents, whereas recent advances in reinforcement learning Guo et al. (2025); Yu et al. (2025); Dong et al. (2025) introduce end-to-end optimization that better unlocks LLMs' capabilities with agentic RL. DeepResearcher Zheng et al. (2025) exemplifies this by employing a multi-agent system with reinforcement learning to promote iterative self-reflection and improved planning in web environments. Furthermore, early-stage trajectory learning Zhang et al. (2025a) demonstrates that strategically curated exploratory experiences significantly enhance agent generalization and tool-use robustness, providing insights into how training dynamics influence agent capabilities.

However, all the work mentioned above ignore one significant question: How to synthesize high-quality data and design training pipelines to fine-tune open source language models to achieve SOTA deep research performance as the close-source models or agent frameworks. Furthermore, effectively synthesizing such data requires simulating the entire research process—from initial query planning to multi-step knowledge retrieval and final synthesis—capturing both successful and instructive failure trajectories. By explicitly modeling the iterative and exploratory essence of deep research, we can generate training corpora that teach models not just what to think, but how to think through a problem. Successfully addressing this gap would democratize access to powerful research assistants, breaking the current dependency on proprietary APIs and closed ecosystems. Ultimately, it paves the way for a new generation of transparent, adaptable, and verifiable open-source models capable of autonomous scientific discovery and rigorous evidence-based reasoning.

## 3 METHOD

Our objective is to develop an end-to-end, open-source model for complex research tasks that minimizes reliance on intricate prompt engineering and hand-designed workflows. We propose a three-stage training pipeline: (1) Design a power and generation deep research report generation workflow and synthesize high-quality trajectory data generation for Supervised Fine-Tuning (SFT), (2) Conduct SFT on an open source language model to empower it to learn the research process, and (3) Conduct reinforcement learning to further enhance the model's tool-use deep research report generation capacity.

### 3.1 HIGH-QUALITY TRAJECTORY GENERATION FOR SFT

The foundation of our approach is a meticulously curated dataset of deep research tool-integrated reasoning trajectories. The process involves two key steps: data synthesize and trajectory synthesis.

**Workflow Introduction** As depicted in Figure 1, we designed a divide-and-conquer report generation workflow. A given query is first decomposed into multiple subqueries, and different models independently conduct multiple rounds of tool-integrated reasoning to produce subquery reports. These intermediate reports are then aggregated by a dedicated model to form the final answer. Meanwhile, the trajectories underlying all subreports are collected and merged into a single SFT reasoning trace. Inspird by pervious work Shi et al. (2025), we introduce LLMs to synthesize thousands of high-quality queries from hundreds of domain, enabling large-scale query construction, trajectory generation, and subsequent quality filtering.

**Query Synthesis** To construct a high-quality corpus for supervised fine-tuning, we curated a seed set of 5,000 queries derived from both mature open-source datasets (including Zhihu-KOL wangrui6 (2024), WideSearch ByteDance-Seed (2024), and ELI5 sentence-transformers (2022)) and LLM-synthesized topics covering under-represented domains. These queries were specifically selected for their open-ended nature

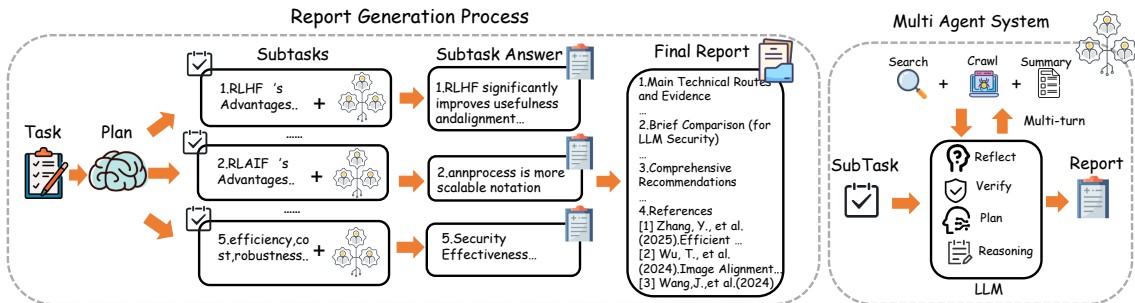

Figure 1: The report generation process of O-Researcher. A query is broken down into multiple sub queries, which are then independently and parallelized by different agents through tool-integrated reasoning to generate sub query reports. These sub query reports are then aggregated through summarizer agent to generate the final report. All traces and reports of different sub-queries are concated as the supervised-training data for this query.

and high complexity. Following the initial generation by our agentic workflow, we applied rigorous filtering to remove low-quality or short-path trajectories. This process ultimately yielded 3,500+ premium instruction-response pairs, which serve as the foundation for our SFT stage.

**Trajectory Synthesis** To construct high-fidelity training data, we employ a diversity-driven synthesis strategy Li et al. (2025c); Chen et al. (2025) followed by a rigorous, multi-stage rejective sampling pipeline. For each curated query, our agentic framework—comprising a planner, tool-user, and summarizer—engages in multi-step *Thought–Tool–Observation* interactions Qin et al. (2025). To capture diverse reasoning patterns, we implement a **Triple Trajectory Sampling** strategy, generating three distinct candidate trajectories for every query. A fusion model then consolidates the decomposed sub-reports into a coherent final answer accompanied by its reasoning trace.

**Quality Assurance Pipeline** To ensure the generation of high-fidelity training data, we have designed a comprehensive, multi-stage rejective sampling pipeline. This pipeline acts as a funnel, starting with a diverse set of candidates and progressively filtering them based on increasingly sophisticated criteria, organized into the following stages:

- **Model-Based Semantic Filtering** Trajectories that survive the deterministic checks proceed to semantic evaluation by a Qwen3-based LLM-as-a-Judge. This model assesses higher-order qualities that rules cannot capture, including logical coherence, the relevance of tool usage, and the evidential grounding of the final answer.

- **Human-in-the-Loop Verification** As the final layer of validation, the highest-rated trajectories undergo topic-stratified human spot-checking. If a sample is flagged as low-quality, it triggers a regeneration loop, where the original query is re-processed until a valid trajectory is produced and verified.

This exhaustive funneling approach ensures that only deep-research trajectories meeting our highest standards are retained for the subsequent SFT and RL phases. Further details are provided in the Appendix.

**Structured Data Representation** To transform raw interaction data into effective training signals, we serialize the trajectories using a coherent XML-style schema. The training sequences are structured to

explicitly expose the model's step-wise reasoning and tool-use behaviors, organized into the following tag categories:

- **Workflow Control Tags:** Define the high-level structure of the problem-solving process.
    - `<subtask_list>`: Decomposes the main query into a sequence of manageable sub-problems.
    - `<subtask>`: Marks the beginning of the execution for a specific sub-problem.
- **Cognitive Tags:** Encapsulate the internal reasoning and planning processes.
    - `<think>`: Contains the internal reasoning monologue, analysis, and strategic thinking before an action is taken.
    - `<plan>`: Outlines a concrete, step-by-step action plan derived from the preceding thought process.
- **Action Tags:** Represent executable tool calls that the model learns to invoke to gather external information.
    - `<web_search>`: Invokes the web search tool with a set of queries.
    - `<crawl_page>`: Invokes the web crawler tool with specific URLs.
- **Feedback Tag:** Contains the raw output returned by a tool after an action is executed.
    - `<observation>`: Captures the direct results from a tool call (e.g., search snippets, webpage content).
- **Response Tags:** Mark the synthesized conclusions at different stages of the process.
    - `<subtask_answer>`: Provides the conclusive answer for a single subtask.
    - `<suggested_answer>`: Marks the final, comprehensive report that consolidates all subtask answers.

This format (examples in Appendix B) forces the model to adhere to a structured *Thinking–Acting–Observing–Answering* loop, which is critical for cultivating robust and verifiable research capabilities.

## 3.2    REINFORCEMENT LEARNING FROM AI FEEDBACK (RLAIF)

To further enhance the model's capability to produce high-quality, novel, and comprehensive research reports, we implemented a reinforcement learning stage utilizing Proximal Policy Optimization (PPO) (Schulman et al., 2017).

**Preference Data Curation.**    As depicted in Figure 2, We began by synthetically generating a diverse set of research questions across multiple domains using an auxiliary large language model (LLM). To construct a preference dataset that is both challenging and informative for reinforcement learning, we filtered these questions based on the performance variance of our supervised fine-tuned (SFT) model. Specifically, for each question, we generated eight distinct responses and evaluated them. Questions that resulted in consistently high scores (indicating trivial difficulty) or consistently low scores (indicating intractable difficulty) were discarded. This filtering approach isolates queries within a "sweet spot" of difficulty, thereby maximizing the learning signal for the policy model during training.

**Reward Function Design.**    Our reward function is designed to balance report quality, tool utilization efficiency, and format compliance. It is formulated as a weighted combination of three primary components:

$$R = w_1 R_{\text{base}} + w_2 R_{\text{tool}} + w_3 R_{\text{format}}. \tag{1}$$

We set $w_1 = 0.9$ and $w_2 = 0.1$, placing stronger emphasis on high-quality report generation while still encouraging appropriate tool-use behavior.

Figure 2: The deep research model training process of our work, which has SFT stage and RL stage.

**Base Quality Reward** ($R_{\text{base}}$). The base quality reward is obtained from an LLM-as-a-Judge that evaluates each (question, generated report) pair along four dimensions: comprehensiveness, insight, instruction-following, and readability. Each dimension is computed via a weighted sum of its criteria, and $R_{\text{base}}$ is the average across dimensions.

**Tool-Usage Reward** ($R_{\text{tool}}$). To encourage appropriate evidence collection, we define

$$N_{\text{calls}} = \min(\texttt{web\_search}, \texttt{crawl\_page}), \qquad N_{\text{min}} = 2,\ N_{\text{max}} = 8.$$

The tool-usage reward is:

$$R_{\text{tool}} = \begin{cases} 0, & N_{\text{calls}} < N_{\text{min}}, \\ -1, & N_{\text{calls}} > N_{\text{max}}, \\ \frac{N_{\text{calls}} - N_{\text{min}}}{N_{\text{max}} - N_{\text{min}}}, & \text{otherwise.} \end{cases}$$

This design rewards reasonable tool usage while penalizing both insufficient and excessive invocation.

**Formatting Reward** ($R_{\text{format}}$). The formatting reward enforces structural correctness. It verifies two strict conditions: (1) all XML-style tags must be symmetrically closed; (2) the output must contain a `<suggested_answer>` tag. Violations of either condition yield zero formatting reward.

**Final Reward.** The final composite reward is normalized to $[0, 1]$ to provide a stable training signal.

During PPO training, we observed several notable trends:

- a steady increase in average response length, suggesting that the model learned to generate more detailed reports;

- a gradual rise in web-search calls and a larger increase in crawl-page calls, indicating deeper evidence gathering;

- an initial sharp decline in policy entropy followed by a gradual rebound, reflecting early convergence to effective strategies before exploring more nuanced behaviors.

## 4 EXPERIMENT

### 4.1 EXPERIMENTAL SETUP

**Benchmark.** We evaluate all models on the validation set of the Deep-Research-Bench(). This benchmark comprises 100 doctoral-level research tasks, designed to test advanced reasoning and information synthesis capabilities. The tasks are distributed across four diverse and knowledge-intensive domains: Science & Technology, Finance & Business, Software Engineering, and Others.

**Baselines.** To establish a robust comparison, we include several state-of-the-art systems as baselines. These are categorized into three groups:

- **Search-enhanced LLMs:** We evaluate representative commercial search-integrated large language models. From OpenAI, we consider O3, GPT-4.1, and GPT-5 (OpenAI, 2025). We also include Gemini-2.5 family (Pro and Flash) (Google DeepMind, 2025), Perplexity Sonar and Sonar-Pro (Perplexity AI, 2025b), and Kimi-k2 (Moonshot AI, 2025). These models represent the state-of-the-art in standard retrieval-augmented generation.

- **Proprietary Deep Research Agents:** We compare against advanced agentic systems designed for multi-step, deep information retrieval. This category includes OpenAI Deep Research (OpenAI, 2025), Gemini-2.5-Pro Deep Research (Citron, 2024), Perplexity Deep Research (Perplexity AI, 2025a), and Grok Deeper Search by xAI (). Additionally, we include recent agent frameworks such as MiroFlow (Team, 2025) and OAgents (Zhu et al., 2025) to benchmark complex reasoning capabilities.

- **Open-sourced Deep Research Models:** To assess the performance of O-Researcher, we compare the SFT and RL variants alongside other open-weights deep research models, including Tongyi-Deep Research (DeepResearch et al., 2025) and MiroThinker (MiroMind et al., 2025).

**Impact of Context Length.** To investigate the effect of context length on learning complex research workflows, we conducted experiments training on trajectory data truncated to 32k and 64k tokens. Our findings indicate a substantial performance improvement when scaling from 32k to 64k, suggesting that longer context is crucial for capturing the complete research arc. However, we observed diminishing returns when scaling from 64k to 128k, implying a potential plateau in performance gains from context length alone.

### 4.2 EVALUATION METRICS

To provide a multi-faceted assessment of model performance, Deep-Research-Bench Du et al. (2025) employs two sets of metrics: **RACE** for evaluating the qualitative aspects of the report and **FACT** for quantifying its factual correctness and citation quality. These are detailed as follows:

**RACE (Report Quality):** This metric assesses the overall quality and presentation of the report through four criteria:

- **Comprehensiveness:** The extent to which the report covers all key aspects of the query.
- **Insight/Depth:** The level of analysis, novelty, and depth beyond simple information aggregation.
- **Instruction-Following:** How well the report adheres to any implicit or explicit constraints in the user's query.
- **Readability:** The clarity, structure, and coherence of the writing.

**FACT (Factual Correctness):** This metric focuses on the reliability and accuracy of the report's content, comprising two main items:

Table 1: Overall evaluation results of DeepResearch Bench.**Bold** denotes the highest score in each column for Proprietary LLMs/Deep Research Agents/Open-sourced Deep Research Models.

| Model | RACE | | | | | FACT | |
|-------|---------|-------|-------|-------|-------|---------|--------|
|       | Overall | Comp. | Depth | Inst. | Read. | C. Acc. | E. Cit. |
| Search-enhanced LLMs | | | | | | | |
| Claude-3-7-Sonnet | 40.67 | 38.99 | 37.66 | 45.77 | 41.46 | 93.68 | 32.48 |
| Claude-3-5-Sonnet | 28.48 | 24.82 | 22.82 | 35.12 | 35.08 | **94.04** | 9.78 |
| Perplexity-Sonar-Reasoning-Pro | 40.22 | 37.38 | 36.11 | 45.66 | 44.74 | 39.36 | 8.35 |
| Perplexity-Sonar-Reasoning | 40.18 | 37.14 | 36.73 | 45.15 | 44.35 | 48.67 | 11.34 |
| Perplexity-Sonar-Pro | 38.93 | 36.38 | 34.26 | 44.70 | 43.35 | 78.66 | 14.74 |
| Perplexity-Sonar | 34.54 | 30.95 | 27.51 | 42.33 | 41.60 | 74.42 | 8.67 |
| Gemini-2.5-Pro | 35.12 | 34.06 | 29.79 | 41.67 | 37.16 | 81.81 | **32.88** |
| Gemini-2.5-Flash | 32.39 | 31.63 | 26.73 | 38.82 | 34.48 | 81.92 | 31.08 |
| OpenAI O3 | 43.71 | 42.02 | 38.80 | 50.29 | 45.90 | 5.80 | 1.10 |
| GPT-5 | **46.77** | **45.41** | **44.54** | 50.29 | **47.47** | 37.87 | 12.21 |
| GPT-4.1 | 33.46 | 29.42 | 25.38 | 42.33 | 40.77 | 87.83 | 4.42 |
| Kimi-K2 | 44.47 | 42.78 | 39.65 | **50.82** | 46.00 | 4.55 | 0.27 |
| Deep Research Agents | | | | | | | |
| Grok Deeper Search | 40.24 | 37.97 | 35.37 | 46.30 | 44.05 | 83.59 | 8.15 |
| Perplexity Deep Research | 42.25 | 40.69 | 39.39 | 46.40 | 44.28 | **90.24** | 31.26 |
| Gemini-2.5-Pro Deep Research | 48.88 | 48.53 | 48.50 | 49.18 | **49.44** | 81.44 | **111.21** |
| OpenAI Deep Research | 46.98 | 46.87 | 45.25 | 49.27 | 47.14 | 77.96 | 40.79 |
| MiroFlow | 44.87 | 44.57 | 49.30 | 45.45 | 46.11 | - | - |
| OAgents | **50.76** | **50.39** | **51.20** | **50.32** | 49.41 | 33.97 | 12.56 |
| Open-sourced Deep Research Models | | | | | | | |
| O-Researcher-SFT | 46.24 | 44.41 | 46.84 | 46.79 | 46.76 | 77.70 | 22.63 |
| O-Researcher-RL | 48.48 | 47.32 | 49.54 | 48.64 | 47.58 | **81.30** | 26.01 |

- **Citation Accuracy:** Whether the claims in the text are accurately supported by the provided citations.

- **Effective Citations:** The relevance and quality of the sources cited to support the report's claims.

## 4.3 PERFORMANCE COMPARISON

The main performance comparison, detailed in Table 1, reveals a clear hierarchy across different model categories. Our proposed methods, particularly the OAgents framework (50.76) and the distilled O-Researcher-RL model (48.48), establish a new state-of-the-art in overall performance, significantly outperforming both general-purpose LLMs and specialized commercial systems.

Among the General LLMs, GPT-5 emerges as the strongest baseline with an overall score of 46.77. While Kimi shows competitive performance in instruction following (50.82), its extremely low citation validity rate (0.06) highlights a critical weakness in factual grounding, a common issue for models not optimized for research tasks.

In the Deep Research Frameworks category, commercial systems like O3 Deep Research and Sonar Research demonstrate their primary strength in citation quality, achieving excellent validity rates of 0.87 and 0.83, respectively. This underscores the value of specialized architectures for enhancing factual trustworthiness.

However, our O-Researcher-RL model surpasses them in overall capability (48.48 vs. 43.50) and excels in generating a high volume of relevant information, achieving the highest total citations (81.30).

A key insight is the trade-off between citation quantity and quality. While our SFT/RL models produce a vast number of citations, their validity rates (0.29 and 0.32) are more aligned with powerful raw models like GPT-5 (0.32) than with the hyper-specialized commercial frameworks. Our teacher system, OAgents, achieves the best balance, leading in overall quality, comprehensiveness (50.39), and insight (51.20).

Table 2: Evaluations on DeepResearchGym.

| Artifacts | Relevance(KPR) | Relevance(KPC) | Faithfulness(Recall) | Quality(Clarity) | Quality(Insight) |
|---|---|---|---|---|---|
| Search-enhanced LLMs | | | | | |
| Kimi-K2-Thinking | 64.32 | 1.28 | 10.56 | 97.4 | 82.8 |
| MiniMax M2 | 37.56 | **0.91** | 5.81 | 52.6 | 43.1 |
| Deep Research Agents | | | | | |
| Sonar-Deep-Research | 66.14 | 1.82 | 30.56 | 96.3 | 86.4 |
| O3-Deep-Research | 62.4 | 1.56 | 33.46 | 92.8 | 76.8 |
| MiroFlow | **80.55** | 1.83 | 25.93 | 96.1 | 95.4 |
| OAgents | 80.17 | 1.16 | 35.65 | **99.5** | **99.7** |
| Open-sourced Deep Research Models | | | | | |
| Tongyi-DeepResearch-30B-A3B | 67.33 | 1.8 | - | 95.90 | 79.1 |
| O-Researcher-32B | 68.35 | 0.93 | 32.01 | 95.9 | 91.1 |
| O-Researcher-72B | 77.28 | 1.74 | **51.45** | 100.00 | 99.3 |

## 4.4 EVALUATIONS ON DEEPRESEARCHGYM

We also evaluate our models on DeepResearchGym Coelho et al. (2025), with 100 instances sampled from the official dataset and compared it to commercial retrieval APIs.While OAgents did not achieve the highest Relevance (KPR) score (due to some fluctuations), it outperforms other DeepResearch Agents in all other metrics. This shows its strength in citation relevance and factual accuracy.In the Deep Research Agents category, O-Researcher-72B leads in generating higher-quality, more relevant research compared to Sonar-Deep-Research and O3-Deep-Research.

## 5 DISCUSSION

**Why we need a divide-and-conquer workflow?** To quantitatively assess the impact of our proposed methodology, we compare the performance of GPT-5 with and without the divide-and-conquer workflow. As illustrated in Table 3, employing the divide-and-conquer strategy yields a substantial improvement across all evaluated metrics. The overall score increases significantly from 0.43 to 0.50. More specifically, we observe notable gains in Comprehensiveness ($0.41 \rightarrow 0.50$) and Insight ($0.39 \rightarrow 0.49$), which are critical dimensions for deep research tasks. This demonstrates that decomposing a complex problem into manageable sub-tasks enables the model to conduct more thorough investigation and generate deeper analysis. Even metrics with already high baseline performance, such as Instruction Following and Readability, see further improvements. These results strongly validate the necessity of a structured workflow for handling complex research queries, as it provides a systematic framework that guides the model to outperform its prompting counterpart consistently.

**The steps concerning the workflow.** What is the optimal step number for generating the workflow? The step number is important since it determines the length of the training data, which directly controls the training cost and the training performance. In this work, we try two different workflow steps: 5 and 10. As illustrated in Table 4, The results indicate that the 10-step workflow consistently outperforms the 5-step

Table 3: The comparison between GPT-5 with and without divide-and-conquer workflows.

| Workflows | Overall | Comp. | Ins. | Inst. | Read. |
|---|---|---|---|---|---|
| Regular Workflow | 42.92 | 40.59 | 38.58 | 48.05 | 46.88 |
| **Divide-and-Conquer** | **49.60** | **49.61** | **48.69** | **50.58** | **50.32** |

approach across all evaluation metrics, achieving a higher overall score (0.496 vs. 0.488). Notably, the 10-step workflow shows the most significant improvement in Comprehensiveness. This suggests that a more fine-grained decomposition of tasks enables a more thorough exploration of the research problem. Therefore, we identify the 10-step workflow as the optimal choice, offering the best balance between performance gains and computational expense. When we increase the step number from 10 to 20,there is no obvious performance gain but much longer retrieval context. Therefore, we choose 10 as our step number.

Table 4: Impact of Reasoning Steps. Performance on the OAgents framework

| Step Number | Overall | Comp. | Ins. | Inst. | Read. |
|---|---|---|---|---|---|
| 5 Steps | 48.80 | 47.89 | 48.31 | 49.81 | 49.91 |
| 10 Steps | 49.61 | 49.60 | 48.71 | **50.62** | **50.31** |
| 20 Steps | **50.76** | **50.39** | **51.20** | 50.32 | 49.41 |

Our ablation studies reveal two critical factors for enhancing performance in deep research tasks. As shown in Table Table 4, adopting a **Divide-and-Conquer** workflow significantly boosts the overall score of GPT-5 from 42.92 to 49.60. This highlights the importance of structured decomposition for complex queries. Furthermore, Table 4 demonstrates a clear correlation between the number of reasoning steps and performance within OAgents framework. Increasing the steps from 5 to 20 elevates the overall score from 48.80 to a peak of 50.76. This suggests that allowing the model more "thinking time" through a greater number of steps is essential for achieving state-of-the-art results. Both findings underscore that advanced agentic workflows, rather than raw model capability alone, are key to unlocking top-tier performance on these challenging benchmarks.

## 6    CONCLUSION

In this work, we introduced a novel multi-agent workflow for the automated synthesis of end-to-end deep research data. By structuring collaboration among multiple LLM agents to decompose, debate, and verify complex tasks, this system provides a scalable pipeline for generating high-quality training corpora. Leveraging this synthesized data, we designed a two-stage training strategy that combines supervised fine-tuning with a novel reinforcement learning method to further refine model performance. Extensive experiments on a major deep research benchmark demonstrate that our framework empowers open-source models of various sizes to achieve new state-of-the-art performance, significantly closing the gap with leading closed-source models and establishing new benchmarks in the open-source domain.

## 7 ETHICAL STATEMENT

This submission adheres to the ICLR Code of Ethics and has been reviewed by all authors to ensure compliance with ethical guidelines for research conduct. No sensitive personal information (e.g., names, contact details, or identifiable biometrics) is retained or analyzed in the study. Second, regarding potential impact: This research focuses on agent models for deep research tasks, and we have conducted a preliminary assessment of its potential risks. We confirm that the methodology and findings do not enable harmful applications. Third, regarding conflicts of interest: All authors declare no financial or non-financial conflicts of interest related to this work. We affirm that all experimental results are reported truthfully, without fabrication or selective reporting, and that all authors have contributed substantially to the work in compliance with research integrity standards.

## 8 REPRODUCIBILITY STATEMENT

The experimental setup and reproducibility conditions are detailed in their respective sections throughout the paper. We will open-source our code to facilitate replication of our results.

## 9 LLM USAGE STATEMENT

In this submission, LLMs are used solely as auxiliary tools for writing polishing, with no LLM deemed a contributor to the work. All authors retain full responsibility for the content, accuracy and integrity of the submission. Specifically, GPT-5 is used to refine the paper's writing clarity and check for grammatical consistency in the manuscript. However, all core content, including the description of methods, presentation of experimental results, and formulation of conclusions, is independently drafted, reviewed, and revised by the authors. No LLM-generated text is included in the submission without a thorough manual verification, and all claims regarding model performance are validated by the authors through experimental testing or mathematical proof.

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

## A  OUR WORKFLOW DESIGN

---

**User Query**

Investigate the future evolution of global volcano early
warning systems.

---

`<plan>`

To comprehensively investigate the "future evolution of
global volcano early warning systems," I will adopt a 5-step
research plan, moving from foundational technologies to
strategic implementation
1.How will future global volcano early warning systems evolve
in terms of technology, governance, and application?
2:What are the key drivers and obstacles for these
technologies, and what is the expected timeline?
3:Analyze existing systems as case studies to ground the
future predictions.
4:Evaluate the potential economic and societal impact.
5:Formulate policy recommendations based on all findings.
This structured plan ensures all facets of the query are
addressed, leading to the following subtask list.

---

`<subtask_list>`

Subtask1:  How will future global volcano early warning
systems evolve in terms of technology, governance, and
application?
Subtask2:  What are the key drivers and obstacles for these
technologies, and what is the expected timeline?
Subtaskn:  ...

---

       ... (Begin processing Subtask 1) ...

`<subtask>`

How will future global volcano early warning systems evolve
in terms of technology, governance, and application?

---

```
<think>

To address the evolution of volcano early warning systems, I
need to gather specific information across three dimensions:
1.  Technology:  I need to investigate upcoming
high-resolution satellite missions (like NISAR) and the role
of AI/Machine Learning in processing seismic data.  CubeSats
are also a potential area for thermal monitoring.
2.  Governance:  I need to find information on international
frameworks for data sharing, as volcanic ash often affects
multiple countries.
3.  Application:  I need to understand how these technical
advancements translate into actual warning protocols.
I will start by performing a broad web search to identify key
projects, technologies, and international bodies involved in
this evolution.
```

```
<web_search>

query=future of volcano monitoring technology
| NISAR satellite volcano monitoring
| AI in volcanic eruption prediction
| international volcano data sharing governance
| CubeSat constellations for thermal monitoring &num=15
```

```
<observation>

[1] NASA NISAR Mission:  Radar vision for watching Earth's
    changes...  launch scheduled for 2024.
[2] Nature Article:  "Machine Learning predicts eruption time
    with 90% accuracy using seismic amplitude"...
[3] UN-SPIDER: "International Charter on Space and Major
    Disasters" activated for volcanic ash monitoring...
    ...  (Total 15 results received)
```

```
<think>

The web search results have provided several high-authority
sources that appear to contain the detailed information
needed to answer the subtask:
1.A NASA JPL page specifically about the NISAR mission.
2.A Nature article discussing AI in eruption prediction.
3.USGS and UN-SPIDER pages that likely contain details on
governance and international charters.
4.An ESA page about CubeSats.
The search snippets are insufficient for a detailed report;
I need to access the full content of these pages to extract
specific technical specifications (e.g., resolution, revisit
time), specific governance agreement names, and detailed AI
methodologies.
```

```
<crawl_page>

url=https://www.jpl.nasa.gov/missions/nisar/volcanoes
| https://www.nature.com/articles/s41598-023-31234-5
| https://volcanoes.usgs.gov/vsc/movies_home.html
| https://www.un-spider.org/news-and-events/news
```

```
<observation>

> From url1 (NISAR): "L-band and S-band radar...  12-day
revisit cycle...  detects deformation < 1 cm."
> From url2 (Nature):  "The hybrid CNN-LSTM model analyzes
real-time seismic tremor data..."
> From url4 (UN-SPIDER): "The 'Universal Data Access Policy'
was adopted to facilitate cross-border..."
```

```
<subtask_answer>

Technological evolution will be driven by next-generation
satellite systems such as NISAR (12-day revisit, <1cm
precision).  AI-based assessment will automate risk forecasts
using CNN-LSTM models.  Governance is shifting towards open
cross-border data sharing (Universal Data Access Policy)...

                ...  (Process Other Subtasks) ...
```

```
<think>

All subtasks are complete.
- Subtask 1 provided technical specs (NISAR, AI) and
governance protocols.
- Subtask 2 identified cost as a key barrier and set the
timeline to 2030.
I will now synthesize these findings into a comprehensive
"Global Volcano Hazard Monitoring Roadmap," structuring it
with an Introduction, Technical Body, and Governance section,
ensuring proper citations for all data points.
```

```
<suggested_answer>

Global Volcano Hazard Monitoring and Early Warning:  An
Integrated Roadmap.
A tripartite evolution|sensing (NISAR, CubeSat), AI-driven
prediction pipelines, and globally aligned data-sharing
governance|will define the next decade of volcano monitoring
innovation...
```

## B  PROMPT

**Available Functions**

You may only use the following 8 functions to answer the question. Each function must be enclosed within its corresponding tags.

- **subtask_list** Break the main question into independent subtasks. Start with `<subtask_list>` and end with `</subtask_list>`.

- **subtask** Marks the specific subtask being executed. Start with `<subtask>` and end with `</subtask>`.

- **think** Internal reasoning before plan/tool. Start with `<think>` and end with `</think>`.

- **plan** Break the subtask into detailed micro steps. Start with `<plan>` and end with `</plan>`.

- **tool** Invoke an external tool.

- **observation** Holds the tool output.

- **subtask_answer** Provide the intermediate answer for a subtask. Start with `<subtask_answer>` and end with `</subtask_answer>`.

- **suggested_answer** Integrate all subtask answers into the final solution.

**Available Tools**

- **`<web_search>`** Queries separated by `|`, append `&serp_num=N`. Example: `<web_search>AI trends | LLM safety&serp_num=20</web_search>`

- **`<crawl_page>`** Fetch deeper information from URLs.

- You may call `<web_search>` multiple times as needed.

**Tool Usage Guide**

- If retrieved information is irrelevant, refine your queries and repeat `<web_search>`.

- Continue searching until you have high-confidence evidence.

**Trail Notes**

- **Workflow:** Start with `<subtask_list>`. Then repeatedly perform: `think` → `plan` → `tool` → `observation`, until sufficient info is gathered, then generate `<subtask_answer>`.

- **Information Gathering:** Use tools multiple times when necessary.

- **Tag Restrictions:** Special tags must not appear in free text, especially inside `<think>`.

**Function Association Instructions**

- Always begin with `<subtask_list>`.

- Then start the first `<subtask>`.

- Inside each `subtask`, `<think>` must appear before plan or tool.

- Output `<subtask_answer>` once enough information is gathered.

- After all subtasks, output `<suggested_answer>`.

**Answering Tips**

- Final `<suggested_answer>` must be entirely in English.

- Must include: Introduction, Body, Conclusion, References.

- Every key fact must include a citation like `[1]`.

**References Section**
Each reference entry must follow:

- `[Number].  URL { Webpage Title`

## C  CASE STUDY

### C.1  QUESTION

> What are the current mainstream AI alignment technologies? Please briefly compare their performance in terms of safety when handling large-scale language models.

### C.1.1  RESULT FROM GPT-5:

> MAIN TECHNICAL ROUTES AND EVIDENCE
>
> - **Supervised Fine-tuning (SFT, including instruction fine-tuning and self-instruction)**
>   - **Idea:**
>   - **Security Performance:** It can significantly improve "usability/compliance with instructions," but in terms of security, it usually only covers risk types clearly present in the training corpus. Its generalization and adversarial robustness are limited, and it is easily broken by jailbreak prompts. While common as a starting point, it often requires subsequent preference optimization or security filtering. Self-instruction work shows that instruction fine-tuning can improve task compliance and generality but does not solve the problem of adversarial jailbreaks. Systematic safety assessments (like TruthfulQA and HELM) also suggest that SFT alone has significant room for improvement in "factuality/realism" and "avoidance of inappropriate content".
>
> - **Reinforcement Learning from Human Feedback (RLHF)**
>   - **Idea:** Collect human preference comparisons (labeling the outputs of two models as superior or inferior), train a reward model, and then use an RL algorithm like PPO to optimize the policy so that the model's output better aligns with human preferences and safety criteria.
>   - **Security Performance:** InstructGPT showed that RLHF significantly improves usefulness and alignment, with a measurable reduction in toxicity and inappropriate content. Anthropic's "Helpful and Harmless Assistant" also proved that RLHF can improve both targets simultaneously, with a lower incidence of harmful responses compared to an SFT-only baseline. Limitations include high cost (human annotation) and potential for over-rejection or being bypassed by jailbreak attacks if reward modeling is inappropriate.
>
> - **Reinforcement Learning from AI Feedback (RLAIF) and "Constitutional AI (CAI)"**
>   - **Idea:** Use a set of public principles ("constitution") to constrain the model, then use the model itself or an auxiliary model to evaluate preferences, reducing reliance on human annotation. It is essentially AI feedback replacing part of human feedback.
>   - **Security Performance:** Anthropic's CAI shows that principle-driven AI feedback can achieve similar or even better performance than human feedback in "harmlessness," and the annotation process is more scalable. The model's rejections are more consistent and traceable to the principles. This approach is better than pure RLHF in cost and scalability but also faces challenges from jailbreaks.
>
> - **Direct Preference Optimization (DPO)**

- **Idea:** Instead of RL, it uses pairwise preference samples for direct parameter optimization (increasing the log-likelihood of "better" outputs and decreasing that of "worse" ones). It achieves an alignment effect similar to RLHF but with simpler, more stable, and offline training.
    - **Security Performance:** The DPO paper shows it is comparable to or even better than RLHF in multiple preference alignment tasks with lower complexity. For security, its capability depends on whether the preference data covers the "harmlessness" dimension. If so, DPO can inherit these security attributes effectively. However, it is still subject to data distribution limitations and adversarial attacks.

- **Security Classifiers/Guardrails**
    - **Idea:** Add a security classifier and policy filtering (e.g., intercepting self-harm, illegal guidance, hate speech) to the model's input or output. This can also be implemented via security system prompts or policy templates.
    - **Security Performance:** Meta's Llama Guard encodes security policies into a multi-label classifier for auditing inputs/outputs, significantly improving coverage of inappropriate content and easing integration with enterprise policies **?**. These guardrails are effective for known risk categories but face risks of false positives/negatives and need continuous updates.

- **Red Teaming and Adversarial Training**
    - **Idea:** Systematically design jailbreak/adversarial prompts to stress-test the model, using the findings for data augmentation and iterative training to repair vulnerabilities.
    - **Security Performance:** Recent work shows that even LLMs aligned by RLHF/CAI can be jailbroken with a high success rate by automated adversarial methods, demonstrating that existing alignments are not indestructible. Continuous investment in red teaming and defense mechanisms is necessary.

BRIEF COMPARISON (FOR LLM SECURITY)

- **Security Effectiveness (Preventing Inappropriate/Dangerous Outputs)**
    - RLHF and CAI are generally superior to SFT alone, significantly reducing toxicity and providing more consistent rejections.
    - DPO, if using data with security preferences, can achieve a similar level of "harmlessness" as RLHF with simpler training.
    - Security guardrails (like Llama Guard) have strong interception capabilities for known risk types and are a necessary layer for deployment but do not replace the alignment of the model itself.

- **Cost and Scalability**
    - **RLHF:** High cost of human annotation and slow iteration.
    - **CAI/RLAIF:** Relying on AI feedback, the annotation cost is significantly reduced and scalability is better.
    - **DPO:** Simpler, offline training pipeline leads to lower cost; the key is obtaining high-quality preference data.
    - **Guardrails:** Deployment and maintenance costs are manageable, but policies need continuous updates.

- **Robustness and Limitations**

– Existing alignment methods (SFT/RLHF/CAI/DPO) can be bypassed by carefully designed prompts; research has demonstrated cross-model transferable jailbreak attacks. They must be used with red teaming and multi-layered protection.

- **Balance between Over-rejection and Practicality**
  – If RLHF/CAI overemphasizes "harmlessness," it may result in "over-conservatism." A balance must be achieved through careful reward design and principle trade-offs.

- **Authenticity and Factuality (as a security-related dimension)**
  – RLHF (like InstructGPT) outperforms unaligned baselines on truthfulness benchmarks like TruthfulQA. However, "hallucination" is not completely eradicated, requiring further enhancements like search augmentation and fact-checking.

COMPREHENSIVE RECOMMENDATIONS

- A common combination in production is: **SFT as a foundation** + **preference optimization (RLHF/CAI or DPO)** to improve "helpfulness and harmlessness" + **security guardrails** for input/output interception + **continuous red teaming** and adversarial data augmentation.

- If cost and scalability are primary considerations, **CAI/RLAIF** or **DPO** should be prioritized. If the highest quality is the goal, **RLHF** remains a strong choice, provided human annotation resources are available. Regardless of the method, continuous security assessments (e.g., TruthfulQA, HELM) and red teaming are essential to maintain security against real-world threats.

## C.1.2 RESULT FROM O-RESEARCHER:

ABSTRACT

This paper synthesizes reports from multiple subtasks, systematically compares and summarizes the advantages, limitations, and key practical points of mainstream alignment techniques in improving the security of large-scale language models (LLMs), and provides the core principles and representative methods of each technique. Key conclusions include: No single method can comprehensively excel in the dimensions of "usefulness, security, robustness, and cost"; engineering practices often adopt a multi-layer alignment stack (data governance → SFT/instruction fine-tuning → preference alignment → red team/runtime protection) to achieve a compromise. RLHF is effective but costly and lacks robustness. RLAIF significantly reduces costs but is sensitive to the quality of the AI "teacher" and remains fragile against strong adversarial attacks. Policy regularization methods (DPO, etc.) have advantages in sample efficiency but limited jailbreak resilience. This paper provides a structured comparison across multiple dimensions and concludes with practical recommendations for building secure and aligned LLMs.

REPORT STRUCTURE

1. Comparative Framework and Evaluation Dimensions

2. Method Comparison (Item-by-Item)

3. List of Mainstream Alignment Techniques

4. Practical Recommendations and Multi-Layer Alignment Stack

5. Conclusions and Research Gaps

6. References

1. COMPARATIVE FRAMEWORK AND EVALUATION DIMENSIONS

For ease of comparison, the following unified evaluation dimensions are adopted:

- **Direct Safety Effects** (Toxicity, Harmlessness, Bias, Factuality)
- **Usefulness/Helpfulness** (instruction-following, helpfulness metrics)
- **Training efficiency** (data requirements, annotation cost, computation/wall time)
- **Deployment cost** (latency, throughput, model size, inference overhead)
- **Robustness** (red team, jailbreak, hint injection, adversarial examples)
- **Cross-domain/multi-turn dialogue security** (round-cumulative risk, toxicity drift)
- **Governability and auditability** (interpretability, reproducibility)

*(Note: Different studies use different benchmarks. This article cites specific reports to indicate comparability conditions.)*

2. METHOD COMPARISON (ITEM BY ITEM)

2.1 COMPARISON OF RLHF WITH OTHER ALIGNMENT TECHNIQUES

- **Advantages (Security-Oriented)**
  - Directly optimizes to human preferences, typically reducing toxicity and improving instruction compliance.
  - Mature Engineering: Data pipelines, PPO training, and RM training processes have been systematized in the industry.
- **Limitations (Security-Oriented)**
  - Label consistency and cultural bias issues; RM training may amplify biases or be affected by reward gaming.
  - Alignment Tax: Security improvements often come at the expense of downstream task performance.
  - Red team evaluations show that training-period alignment alone cannot completely prevent systematic abuse.
  - High Cost: High-quality human preference labeling is expensive and difficult to scale.
- **Key Comparisons**
  - **vs. DPO:** Simpler and more efficient training with results close to RLHF, but with a risk of bias in out-of-distribution safety.
  - **vs. Constitutional AI (CAI):** Achieves large-scale harmless training via AI self-annotation, but its effectiveness is limited by the accuracy of the constitution.
  - **vs. SFT:** Lower cost and easier to deploy, but with limited safety improvement. Often used as a prerequisite for RLHF/DPO/CAI.

2.2 COMPARISON OF RLAIF WITH OTHER ALIGNMENT TECHNIQUES

- **Advantages (Security-Oriented)**
  - Using an AI to generate preference labels or scores significantly reduces cost and makes alignment scalable, approaching the effect of RLHF.

- d-RLAIF (Online LLM Scoring) eliminates the RM training step, simplifying the process.
- Frameworks like SRPO can improve robustness under unstable preferences.

- **Limitations (Security-Oriented)**
  - Highly sensitive to the quality of the "evaluator/teacher" model.
  - Robustness to red team/jailbreak evaluation is not significantly better than other training-period alignment methods.
  - AI-generated labels have poor interpretability and auditability.

- **Key Comparisons**
  - **vs. RLHF:** Alignment quality is comparable but at a lower cost; however, it requires a strong teacher model.
  - **vs. DPO:** d-RLAIF avoids training an RM but online invocation incurs deployment costs; C-DPO provides a compromise.
  - **vs. SFT:** RLAIF can impose more explicit preference targets but relies on AI commenting quality.

### 2.3 Four-dimensional comparison of RLAIF / RLHF / Policy Regularization

- **Training efficiency:** RLAIF $>$ DPO $\approx$ RLHF.
- **Deployment cost:** DPO $\approx$ SFT is lowest; RLAIF increases cost if online scoring is used.
- **Robustness (Toxicity/Jailbreak):** All methods are still vulnerable to systematic jailbreaks. DPO needs specific improvements for OOD security.
- **Multi-turn Conversation Security:** Multi-turn toxicity drift is an open problem for all methods, requiring specific multi-turn alignment data.

### 2.4 Comparison of Policy Regularization (CLIP/CoT) with Other Alignment Methods

- **Advantages**
  - High sample and data efficiency; can significantly reduce online sampling computation.
  - CoT alignment can improve inference robustness and cross-task capability.

- **Limitations**
  - External rewards (like CLIP) are not entirely consistent with text security goals, causing side effects like bias amplification.
  - Limited performance in automated red team/jailbreak evaluations.

## 3. List of Mainstream Alignment Techniques

- **A. Instruction Fine-tuning and Preference Alignment:** Core methods to align LLMs to user intent.
  - *Methods:* InstructGPT (SFT$\rightarrow$RM$\rightarrow$PPO), DPO, Constitutional AI.
- **B. RLHF and its Safe Variations:** Introduce safety constraints into the RLHF process.
  - *Methods:* Safe RLHF, Cost-constrained RLHF, PMLR.
- **C. Data and Post-Training Layers:** Reduce risks via data construction or post-training filtering.

- *Methods:* Toxicity fine-tuning, bias classifiers, debiased datasets.
- **D. Security During Inference (Guardrails):** Apply rules and filtering to the decoding chain.
  - *Methods:* Guardrail frameworks, restricted decoding, online toxicity classifiers.
- **E. Automated/Proxy Alignment Paradigm:** Replace human feedback with model self-criticism.
  - *Methods:* H1–H5, RLAIF, IDA.
- **F. Multimodal and Robust Security:** Attack/defense systems for adversarial prompts.
  - *Reviews:* PromptBench and JailbreakBench.

4. PRACTICAL RECOMMENDATIONS AND MULTI-LAYER ALIGNMENT STACK

- **Multi-layer alignment stack:**
  1. **Data Governance:** Data depollution, debiasing, and adversarial pollution detection.
  2. **Training Phase:** SFT $\rightarrow$ Preference Alignment (RLHF / DPO / RLAIF / CAI), adding safety constraints as needed.
  3. **Model Evaluation:** Multi-benchmark evaluation (helpfulness, toxicity, jailbreak) plus red team testing.
  4. **Runtime Protection:** Guardrails, decoding filtering, content review, and continuous online monitoring.
- **Method Combinations and Trade-offs:** Prioritize DPO/CAI/RLAIF for computational constraints; prioritize RLHF for human preference consistency. Employ multi-layered protection in safety-critical scenarios.
- **Risk Management and Governance:** Establish model cards, red team reports, versioning, and external compliance reviews.

5. CONCLUSIONS AND RESEARCH GAPS

- **Key Conclusions**
  - No single method excels across "usefulness—safety—robustness—cost." Engineering often employs multi-layered, hybrid approaches.
  - RLAIF and DPO offer scalable pathways, but all methods still have shortcomings against strong adversarial attacks and multi-round toxic drift.
  - Automated alignment (H1–H5) is an important future direction but faces challenges like reward hacking and error accumulation.
- **Research Gaps and Suggested Directions**
  - Data and benchmarks for multi-round toxic drift are still insufficient; continuous investment in red teams and benchmarks is needed.
  - Provable trade-offs between security, capability, and computation remain unclear. Robust alignment methods across languages and cultures are lacking.
  - Reward model robustness, reward hacking detection, and theoretical guarantees for RLHF/RLAIF are important research directions.

