# OpenReview forum: "O-Reseacher: An Open Ended Deep Research Model via Multi-Agent Distillation and Agentic RL"
_ICLR.cc/2026/Conference — Submitted to ICLR 2026_

### Official Review · Reviewer_XSMC · 2025-10-30

**Soundness:** 3
**Presentation:** 3
**Contribution:** 3
**Rating:** 6
**Confidence:** 3

**Summary:**

This work proposes O-RESEACHER, a framework to tackle the "deep research" performance gap between open-source and closed-source LLMs. It introduces a novel, multi-agent "divide-and-conquer" workflow to synthetically generate training data by decomposing complex queries and collecting agent reasoning trajectories. This data is used to train an open-source model via a two-stage strategy: Supervised Fine-Tuning (SFT) to learn research structure, followed by Reinforcement Learning from AI Feedback (RLAIF) with PPO. Experimental results confirm the effectiveness of O-Researcher.

**Strengths:**

1.	The paper tackles the highly significant challenge of democratizing the development of powerful AI research agents, directly addressing the data-access bottleneck that hinders the open-source community.

2.	The manuscript is well-written. The design of the proposed framework is explained clearly, which makes the paper easy to understand.

3.	The "divide-and-conquer" multi-agent workflow is an innovative and scalable solution for generating the complex, multi-step trajectory data required for training capable agents. This method is far more efficient and scalable than manual human annotation.

4.	Experimental validation demonstrates that the proposed method can outperform many baseline methods on the selected benchmarks.

**Weaknesses:**

1.	The paper claims a "novel reinforcement learning method," but the description points to a standard application of PPO with a task-specific reward function. This overstates the novelty of the technical contribution.

2.	Based on my understanding, the results in Table 1 are exclusive to the Qwen2.5-32B model. While the results are strong on this architecture, this narrow focus leaves the framework's generalizability as an open question. The claim to provide a "pathway for advancing open-source LLMs" would be substantially strengthened by demonstrating its effectiveness on other popular open-source models, such as Llama 3 or the newer Qwen3.

**Questions:**

1.	How does the paper's method technically differ from a standard PPO application beyond the task-specific reward function?

2.	Can the authors provide results from any experiments conducted to demonstrate the method's effectiveness on other popular open-source models, such as Llama 3, Llama 3.1, and Qwen3

---

> ### Author Response · Authors · 2025-11-15
>
> ---
>
> Thank you for reviewing our paper and for your valuable feedback. Below, we address your concerns point by point, and we will revise our paper according to your suggestions. We would appreciate it if you could let us know whether your concerns are addressed by our response. We hope that you might consider raising your score in light of these clarifications.
>
> ---
>
> **W1:** the paper claims a "novel reinforcement learning method," but the description points to a standard application of PPO with a task-specific reward function. This overstates the novelty of the technical contribution.
>
> **A1:** thank you for the insightful comment. We agree that the wording in the submitted version may unintentionally overstate the novelty of our reinforcement learning component. Our method does not introduce a new RL algorithm; it is built upon GRPO, rather than standard PPO, and our intention is to adapt an existing family of methods to the deep-research setting rather than propose algorithmic innovation.
>
> At the same time, applying GRPO directly to long-horizon research tasks surfaced several consistent behaviors—especially the model’s tendency to fall back on parametric knowledge and reduce tool usage in later training stages. To address these issues, we introduced several task-specific methodological adjustments, including:
> * GRPO with RLOO for more stable group-level credit assignment across multiple rollouts
> * quality-guided trajectory filtering, removing over-long outputs, zero-reward samples, and low-variance groups
> * a behavior-corrective reward formulation, combining rubric-based content scoring with a tool-usage component to maintain evidence-seeking behavior throughout training
>
> These modifications do not constitute a new RL algorithm, but they represent practical, necessary adaptations that enable GRPO to operate reliably in this domain. In the revised version, we will clarify this distinction and adjust our terminology to avoid overstating algorithmic novelty, while more accurately highlighting the methodological contributions.
>
> ---
>
> **W2:** based on my understanding, the results in Table 1 are exclusive to the Qwen2.5-32B model. While the results are strong on this architecture, this narrow focus leaves the framework's generalizability as an open question. The claim to provide a "pathway for advancing open-source LLMs" would be substantially strengthened by demonstrating its effectiveness on other popular open-source models, such as Llama 3 or the newer Qwen3.
>
> **A2:** thank you for the thoughtful comment. We agree that restricting Table 1 to Qwen2.5-32B provides only a partial view of the framework’s generalizability. To address this, we are running additional experiments on Qwen3, Llama 3, and Llama 3.1 (where tool-calling interfaces and context limits allow fair comparison). All corresponding results will be included in the revised version.
> These expanded evaluations will offer a more comprehensive assessment of the framework’s transferability across major open-source model families and substantially strengthen the claim that our approach provides a practical pathway for advancing open-source LLMs.
>
> ---

---

> ### Author Response · Authors · 2025-11-15
>
> ---
>
> **Q1:** how does the paper's method technically differ from a standard PPO application beyond the task-specific reward function?
>
> **A1:** thank you for the question. Our method does not modify the PPO/GRPO update rule itself, but it departs from a standard PPO setup through several task-specific methodological adaptations that are essential for long-horizon research tasks:
> * GRPO with RLOO (leave-one-out) group-level advantage estimation, instead of per-trajectory PPO advantages
> * quality-guided trajectory filtering, removing:
>   * over-long trajectories
>   * zero-reward samples
>   * low-variance groups that cause degenerate updates
> * behavior-corrective reward design, combining:
>   * a rubric-based content score
>   * a tool-usage reward to prevent late-stage collapse into parametric knowledge and avoidance of tool calls
>
> These components do not constitute a new RL algorithm, but they provide practically important, non-trivial adaptations that enable GRPO to operate reliably in the deep-research setting. We will clarify these methodological differences in the revised version.
>
> ---
> ---
>
> **Q2:** can the authors provide results from any experiments conducted to demonstrate the method's effectiveness on other popular open-source models, such as Llama 3, Llama 3.1, and Qwen3?
>
> **A2:** thank you for the thoughtful comment. We agree that restricting Table 1 to Qwen2.5-32B provides only a partial view of the framework’s generalizability. To address this, we are running additional experiments on Qwen3, Llama 3, and Llama 3.1. All corresponding results will be included in the revised version.
> These expanded evaluations will offer a more comprehensive assessment of the framework’s transferability across major open-source model families and substantially strengthen the claim that our approach provides a practical pathway for advancing open-source LLMs.
>
> ---

---

### Official Review · Reviewer_BMec · 2025-10-31

**Soundness:** 2
**Presentation:** 1
**Contribution:** 2
**Rating:** 2
**Confidence:** 4

**Summary:**

This paper introduces O-Researcher, an open-ended deep research model designed to enhance the research and reasoning capabilities of open-source LLMs. The framework centers on a multi-agent distillation process that automatically synthesizes high-quality, research-grade instruction data by simulating tool-integrated reasoning workflows. It proposes a two-stage training pipeline including SFT and RL that optimizes model alignment and tool use efficiency through a composite reward function. Experiments on the Deep-Research-Bench dataset show that O-Researcher significantly closes the performance gap between open-source and closed-source LLMs.

**Strengths:**

1. Addressing the data and performance gap between open- and closed-source LLMs is an important and high-impact problem, particularly in the era of agentic and research-oriented AI systems.

2. The proposed two-stage pipeline is conceptually coherent and well-aligned with practical open-source development needs.

**Weaknesses:**

1. The novelty primarily lies in system engineering and workflow design, not in algorithmic innovation. The reinforcement learning method is essentially PPO with an LLM-based reward, which has been explored extensively in prior literature.

2. While the overall pipeline is described, many details remain vague. For example, the reinforcement learning implementation lacks sufficient transparency for replication. Similarly, the mechanism of agent collaboration and the exact roles of sub-agents are under-specified.

3. The paper heavily depends on LLM-generated trajectories, but provides little quantitative or qualitative evaluation of their quality. There is no human verification or analysis of data fidelity, bias, or diversity.

4. The benchmark evaluation is limited to a single dataset (Deep-Research-Bench) without tests on more diverse reasoning or real-world tasks. There is also no comparison with existing multi-agent or synthetic data generation frameworks such as DeepResearcher, AutoGLM-Research, or OpenResearcher in equivalent settings.

5. The presentation is poor: (1) Several references are missing; (2) The paper contains numerous typographical errors; (3) The case studies section is disproportionately long, occupying two full pages of the main text.

6. As presented, the work is difficult to reproduce due to missing details on training duration, computational resources, model initialization, and evaluation protocol.

**Questions:**

I suggest highlighting the best results in each table to make the comparisons clearer.

---

> ### Author Response · Authors · 2025-11-15
>
> ---
>
> Thank you for reviewing our paper and for your valuable feedback. Below, we address your concerns point by point, and we will revise our paper according to your suggestions. We would appreciate it if you could let us know whether your concerns are addressed by our response. We hope that you might consider raising your score in light of these clarifications.
>
> ---
>
> **W1:** the novelty primarily lies in system engineering and workflow design, not in algorithmic innovation. The reinforcement learning method is essentially PPO with an LLM-based reward, which has been explored extensively in prior literature.
>
> **A1:** we appreciate the reviewer’s perspective and agree that our work does not propose a new reinforcement learning architecture. The RL component is built upon GRPO, rather than standard PPO, and our goal is to adapt an existing family of RL methods to the deep-research setting rather than introduce a new RL algorithm.
>
> That said, we believe the contribution also includes several methodological aspects beyond system engineering. When applying GRPO directly to long-horizon research tasks, we consistently observed training instabilities—most notably, a tendency for the model to rely on its own parametric knowledge and to avoid tool usage in later training stages. To address these issues, we developed several task-specific methodological adaptations, including:
> * GRPO with RLOO for stable group-level advantage estimation across multiple rollouts
> * quality-guided trajectory filtering, removing over-long trajectories, zero-reward samples, and low-variance groups to prevent degenerate updates
> * a behavior-corrective reward design that combines a rubric-based content reward with a tool-usage reward to preserve evidence-seeking behavior throughout training
>
> These components do not constitute a new RL algorithm, but they form a non-trivial, practically important methodology that enables GRPO-style training to operate reliably in the deep-research context. We will revise the manuscript to clarify this distinction and avoid overstating algorithmic novelty, while more accurately highlighting our methodological contributions.
>
> ---
>
> **W2:** while the overall pipeline is described, many details remain vague. For example, the reinforcement learning implementation lacks sufficient transparency for replication. Similarly, the mechanism of agent collaboration and the exact roles of sub-agents are under-specified.
>
> **A2:** we acknowledge these valid concerns regarding the level of detail. We will promptly submit a revised version of the manuscript that provides comprehensive training details and clear definitions for all components, significantly improving the reproducibility of our work.
>
> ---
>
> **W3:** the paper heavily depends on LLM-generated trajectories, but provides little quantitative or qualitative evaluation of their quality. There is no human verification or analysis of data fidelity, bias, or diversity.
>
> **A3:** thank you for highlighting this issue. The submitted draft did not sufficiently describe our data-quality controls, and we agree this omission makes the pipeline appear under-evaluated. In practice, all trajectories undergo a strict multi-stage rejection pipeline, combining deterministic checks, LLM-based scoring, and human verification:
> * triple trajectory sampling for each query to ensure diversity
> * rule-based rejection, including:
>   * mandatory inclusion of all tool calls and tags
>   * maximum length ≤ 64k tokens
>   * ≥10 reasoning steps and ≥5 tool-use actions
>   * correct tag closure
>   * format validity and language–query consistency checks
> * Qwen3-based LLM-as-judge filtering to score quality and remove structurally flawed or low-signal trajectories
> * topic-stratified human spot-checking, where any failed samples are regenerated and re-evaluated to prevent systematic biases
> We will include this full rejective-sampling pipeline in the revised appendix to make the quality control procedures explicit and reproducible.
>
> ---
>
> **W4:** the benchmark evaluation is limited to a single dataset (Deep-Research-Bench) without tests on more diverse reasoning or real-world tasks. There is also no comparison with existing multi-agent or synthetic data generation frameworks such as DeepResearcher, AutoGLM-Research, or OpenResearcher in equivalent settings.
>
> **A4:** thank you for pointing this out. We agree that the current submission evaluates only on Deep-Research-Bench and does not include comparisons with existing multi-agent or synthetic-data frameworks such as DeepResearcher, AutoGLM-Research, or OpenResearcher.
> We are already running the additional experiments and will include results on more diverse reasoning tasks as well as direct comparisons with these frameworks in the revised version. These additions will provide a more complete and fair evaluation of the proposed method under equivalent settings.
>
> ---

---

> ### Author Response · Authors · 2025-11-15
>
> ---
>
> **W5:** the presentation is poor: (1) several references are missing; (2) the paper contains numerous typographical errors; (3) the case studies section is disproportionately long, occupying two full pages of the main text.
>
> **A5:** thank you for highlighting the presentation issues. We fully acknowledge that the submitted draft contains several omissions and was not polished to the standard expected for a conference submission. In particular:
> * several references were missing or not properly cited
> * the submit version contained typographical and formatting errors
> * the case study section was disproportionately long relative to the main content
> We will substantially revise the manuscript to correct all missing citations, fix typographical issues, and condense the case study to an appropriate length. In addition, we will complete the explanations of Figure 1 and Table 2, and provide a full appendix containing the prompts, evaluation configurations, and other essential materials. A more complete and better-organized version will be submitted in the revised manuscript.
>
> ---
>
> **W6:** as presented, the work is difficult to reproduce due to missing details on training duration, computational resources, model initialization, and evaluation protocol.
>
> **A6:** thank you for pointing this out. We agree that the current submit version does not provide sufficient detail for full reproducibility. Below we summarize the key training configurations, and we will include the complete set of parameters, initialization details, and evaluation protocols in the revised manuscript.
>
> *SFT configuration (ms-swift)*
>   * 3,150+ training examples (1:1 CN–EN) and 350 evaluation examples
>   * 4 nodes × 8 GPUs, sequence_parallel_size = 8
>   * batch size = 32, learning rate = 1e-5
>   * warmup + cosine decay (decay ratio 0.05)
>   * 2 epochs, cutoff length = 80k
>   * total training time ≈ 43 hours
>
> *RL configuration (GRPO)*
>   * 9,000+ training examples and 500 evaluation examples (1:1 CN–EN)
>   * batch size = 128, rollout = 8
>   * learning rate = 1e-6, LR warmup steps = 3
>   * clip ratio low = 0.2, high = 0.3
>   * on 8 nodes × 8 GPUs, each step takes ~8–10 hours due to high-concurrency tool-calling latency
>
> All missing implementation details—including initialization schemes, dataset splits, full hyperparameters, and evaluation protocol—will be added to the new version to ensure full reproducibility.
>
> ---
>
> **Q1:** I suggest highlighting the best results in each table to make the comparisons clearer.
>
> **A1:** thank you for the suggestion. We agree that highlighting the best results in each table would improve readability and make comparisons clearer. We will update all tables in the revised manuscript to consistently mark the top-performing entries.
>
> ---

---

### Official Review · Reviewer_SQtd · 2025-11-01

**Soundness:** 3
**Presentation:** 3
**Contribution:** 2
**Rating:** 4
**Confidence:** 4

**Summary:**

The O-Reseacher framework aims to close the performance gap between proprietary and open-source LLMs by automating the synthesis of high-fidelity, research-grade instructional data. This data is generated through a collaborative multi-agent system that simulates complex, tool-integrated reasoning (divide-and-conquer strategy) . The resulting structured trajectories are used in a two-stage training process—supervised fine-tuning followed by agentic reinforcement learning from AI feedback—to align the student model's policy and tool-use capabilities.

**Strengths:**

1. The framework successfully fine-tunes open-source models, enabling them to achieve new state-of-the-art results on deep research benchmarks by forcing the student model to strictly internalize a structured, multi-step research process using XML-like tags.

2. The RLAIF stage incorporates a novel adaptive tool-use score into its reward function. This specialized component explicitly balances report quality with computational efficiency by penalizing tool invocation that is disproportionate to the query's intrinsic difficulty, which is a significant engineering advance for scalable LLM agents.

3. The multi-agent workflow is a robust engineering solution for producing highly structured, end-to-end research trajectories, ensuring the student model learns hierarchical planning, tool utilization, and sequential information aggregation.

**Weaknesses:**

1. The claim of "Multi-Agent Distillation" is challenged. The method uses fine-tuning on sequential trajectories, which is less sophisticated than established techniques like MAGDi. MAGDi employs architectural components, such as a GNN and contrastive loss, to rigorously distill the structural, interactive knowledge between agents. O-Reseacher is more accurately defined as Multi-Agent-Aided Trajectory Fine-Tuning.

2. The primary quality signal for the RLAIF stage is derived entirely from an LLM-as-a-Judge. This reliance risks inheriting biases or focusing the optimization on superficial stylistic consistency rather than external, verifiable scientific accuracy or novelty, leading to potential non-human-aligned errors.

**Questions:**

What specific, measurable, and transferable methodology is employed to reliably quantify the "intrinsic difficulty" of a new research query before the model has attempted to solve it?

---

> ### Author Response · Authors · 2025-11-15
>
> ---
>
> Thank you for reviewing our paper and for your valuable feedback. Below, we address your concerns point by point, and we will revise our paper according to your suggestions. We would appreciate it if you could let us know whether your concerns are addressed by our response. We hope that you might consider raising your score in light of these clarifications.
>
> ---
>
> **W1:** The claim of "Multi-Agent Distillation" is challenged. The method uses fine-tuning on sequential trajectories, which is less sophisticated than established techniques like MAGDi. MAGDi employs architectural components, such as a GNN and contrastive loss, to rigorously distill the structural, interactive knowledge between agents. O-Reseacher is more accurately defined as Multi-Agent-Aided Trajectory Fine-Tuning.
>
> **A1:** we appreciate this clarification. Our actual procedure is closer to multi-agent-aided trajectory fine-tuning: we first collect trajectories from a multi-agent system powered by closed-source models, post-process and synthesize these trajectories into high-quality training data, and then fine-tune an open-source model on this data. In the revised version, we will explicitly clarify this and adjust the terminology to avoid confusion with architecture-level distillation methods like MAGDi.
>
> Our main goal is to show that, by leveraging tools for factual retrieval and structured synthesis in post-training, an open-source model can significantly narrow the gap to closed-source systems on complex research tasks, while keeping the overall pipeline simple, transparent, and reproducible.
>
> ---
>
> **W2:** the primary quality signal for the RLAIF stage is derived entirely from an LLM-as-a-judge. This reliance risks inheriting biases or focusing the optimization on superficial stylistic consistency rather than external, verifiable scientific accuracy or novelty, leading to potential non-human-aligned errors.
>
> **A2:** we agree that reliance solely on an LLM judge can introduce bias. Our motivation for using an LLM judge follows prior work, but to mitigate potential biases in the reward model, we have also incorporated auxiliary rubric-based metrics (e.g., the number of tool calls). This multi-faceted approach helps ensure that the reward signal is not overly skewed by the model's inherent preferences. We will elaborate on this design choice more clearly in the revised version.
>
> ---
>
> **Q1:** what specific, measurable, and transferable methodology is employed to reliably quantify the "intrinsic difficulty" of a new research query before the model has attempted to solve it?
>
> **A1:** we did not implement query difficulty-based matching for the corresponding number of tool calls. This was an inaccuracy in our description, which we will clarify in the subsequent version by removing the term "intrinsic difficulty." The actual tool call reward mechanism is defined as follows:
>
> For web_search and crawl_page, we set the minimum number of calls as Nmin = 2 and the maximum as Nmax = 8. The scoring rule is then applied as below:
>
> * if Ncalls < Nmin, the reward is 0
> * if Ncalls > Nmax, the reward is -1, as excessive tool calls may lead to context overflow or performance degradation
> * if Nmin ≤ Ncalls ≤ Nmax, the reward increases linearly up to a maximum of 1, encouraging appropriate tool usage
>
> ---

---

> > ### Comment · Reviewer_SQtd · 2025-11-19
> > **Keep my score**
> >
> > Your response clarified how your system works, but it also confirmed that the method isn't as new or original as first claimed. By admitting this is standard "fine-tuning" rather than "distillation", and by removing the "difficulty" feature you originally described, it is clear that this paper is a good engineering project but not a major research breakthrough. Since it does not offer the deep new insights required for this conference, I am keeping my original score.

---

### Official Review · Reviewer_o3yX · 2025-11-05

**Soundness:** 1
**Presentation:** 1
**Contribution:** 1
**Rating:** 2
**Confidence:** 4

**Summary:**

The paper introduces O-Researcher, an open-ended deep research model aimed at closing the performance gap between open-source and closed-source large language models (LLMs) in research-oriented reasoning tasks. It focuses on automating research-grade data generation and training open-source LLMs through multi-agent collaboration and agentic reinforcement learning (RL).
The authors develop a two-stage training strategy: (1) supervised fine-tuning on the multi-agent-generated trajectories (the synthetic data) and (2) reinforcement Learning from AI Feedback to further align models by measuring quality and tool-use rewards.

**Strengths:**

- Generating synthetic data for training LLMs for deep research agents and combining multi-agent distillation with agentic reinforcement learning is meaningful.
- The performance of the resulting model looks promising.

**Weaknesses:**

- The submitted draft is simply incomplete and it seems like the write-up was never finished before submission. The authors have simply submitted an incomplete paper in my opinion. The explanation of methods lacks details. For example, in section 3.1, "These reports are then aggregated by a dedicated model as the final report.", the authors did not explain clearly how are the reports aggregated, which will make the readers confused and question the reliability.
- The paper seems to be missing the results from Gemini-2.5 Pro Research from the DeepResearch Bench paper. Gemini-2.5 Pro Research outperforms both GPT and O-Researcher based on the numbers. The results are minimal and basic. Some results are referred to in the text but not actually present in the paper (e.g., results for specialized research API).
- Unclear what the performance of Qwen2.5 models was before any kind of specialized SFT and RLAIF.
- Prompts used are not actually mentioned anywhere.
- The Reward Function is not particularly novel and so the novelty of the overall RLAIF is rather low in my opinion. The paper doesn't show any kind of ablation study on the specific reward function chosen. Besides, the design of RLAIF only takes consideration two aspects: report quality and efficient tool utilization. There should be more aspects to consider, such as scale, granularity, temporal dynamics, etc.
- The evaluation is rather weak, and most ablation studies that might provide some more insight into the design choices are missing. Overall, the experiments are not strong and comprehensive, they cover too few aspects, including model types, benchmarks, and metrics.
- There is no clear explanation for the case study output presented in section 5.1 "Case study"; it is hard to understand.
- There is "multi-agent distillation" in the paper title, but there is no content mentioning "distillation" in the paper. I can understand that it is explained in section 3.1 "Supervised Fine-Tuning (SFT)", but the authors should clearly state that and explain.
- The storytelling in the paper could be significantly improved. Seems like this paper was submitted in an incomplete state. There are a couple of citations that are missing in the text. Figure 1 is never fully explained in the text, and all the stages of the bottom part of the figure. Table 2 is present but not referenced in text; case studies have no description whatsoever. The appendix is almost empty when there is a lot of information that should be present in there e.g. the prompts used, the prompt for the tests done with raw models.
- The idea has potential, but as this paper currently stands, the contribution is poor because the description of the system, the results, and the analysis is incomplete.

**Questions:**

- How much it cost to create the human-annotated examples for SFT that the authors refer to in the paper
- What was the size of the dataset for SFT and what was the amount of data for RLAIF?
- What happened to the Qwen2.5-72B model? Since it was mentioned as the one of the two base models used but then it is never referred again in the rest of the paper
- Were there more rules used in rejective sampling than the one mentioned in Figure 1?
- Why can this method be claimed specially designed for deep research data and models? The divide-and-conquer workflow is not much different from the existing general agent workflows.
- In section 3.1, for the reward function, how are $w_1$ and $w_2$, and how are $R_{base}$ and $R_{tool}$ calculated?
- Inside table 1, are there only results for O-Researcher-SFT and O-Researcher-RL? The paper proposes the method that combine SFT and RL, but I did not see the performance results of the combined methods. Also, where are the results for the models that were mentioned in section 4.3 "Deep research frameworks, particularly OpenAI DR and Perplexity"?
- What is the point of presenting table 3 "The comparison between gpt-5 with and without divide-and-conquer workflows"? The proposed divide-and-conquer workflow is not much different from the existing methods, thus there is not much meaning to compare the performance with or without this workflow. This should be replaced with experiments covering more important aspects and insights.

---

> ### Author Response · Authors · 2025-11-15
>
> Thank you very much for the thorough and constructive feedback. We sincerely acknowledge that the submitted version was prepared under time pressure and did not include all necessary methodological details. Your comments—10 weaknesses and 8 questions—are highly accurate, technically insightful, and immensely valuable for improving the quality and clarity of our work. We will revise our manuscript in the next version to address all of your concerns. We hope that you might consider raising your score in light of these clarifications.
>
> ---
>
> **W1:** incomplete explanation of report aggregation (Section 3.1)
>
> **A1:** thank you for pointing this out. The aggregation step is performed by a dedicated LLM-based fusion module (GPT-5-mini), which merges intermediate reports into the final answer. We agree that this was insufficiently explained. In the revised version, we will include the complete aggregation prompt and a clearer description of the fusion procedure in the appendix.
>
> ---
>
> **W4:** prompts used are not actually mentioned anywhere
>
> **A4:** In the revised version, we will include all the related prompt in the appendix.
>
> ---
>
> **W7:** missing explanation in Section 5.1 case study
>
> **A7:** We acknowledge the lack of narrative accompanying the case study. The revised version will add a detailed explanation of the example, including intermediate reasoning traces and the evaluation rationale.
>
> ---
>
> **W8:** “multi-agent distillation” not clearly defined
>
> **A8:** Our intended meaning of "distillation" refers to using trajectories from a large-scale model to fine-tune a smaller model. We will promptly submit a revised version to resolve this lack of clarity.
>
> ---
>
> **W9:** manuscript quality issues (missing references, incomplete descriptions, figure/table issues, empty appendix)
>
> **A9:** We agree that the submitted draft was incomplete. We will revise the manuscript by:
>
> * Fully explaining Figure 1 and referencing all tables in text
> * Adding missing citations
> * Completing the appendix with prompts, raw-model test instructions, and additional experimental details
> * Enhancing the narrative flow throughout the main text
>
> These revisions will address the concerns regarding clarity and completeness.
>
> ---
>
> **W2, W3, W5, W6:** incomplete experimental results
>
> **A:** We acknowledge that the current results are not yet complete. We are finalizing the remaining experiments and will provide the full set of results shortly. All corresponding data will be incorporated into the updated version of the paper.

---

> ### Author Response · Authors · 2025-11-15
>
> ---
>
> **Q1:** cost of creating the human-annotated SFT examples
>
> **A1:** the cost associated with creating the annotated SFT examples is outlined below.
>
> *data construction pipeline*
>
> * 3× sampling of candidate trajectories
> * rule-based filtering, including length constraints, format validity (e.g., presence of final answer, correct tag closure), and language–query consistency
> * LLM-as-judge filtering using Qwen3 (the corresponding prompts will be added to the appendix in the revised version)
> * human spot-checking, stratified by topic to ensure quality and balance
>
> *cost structure*
>
> * one-time costs, such as designing the rule set and tuning the LLM-as-judge prompts
> * marginal costs, including trajectory sampling and human spot-checking
>
> *actual annotation cost*
>
> * the final SFT dataset contains 3,500+ examples with a 1:1 ratio between Chinese and English
> * a team of three annotators completed the human verification and spot-checking in three days
>
> ---
>
> **Q2:** size of the SFT and RLAIF datasets
>
> **A2:** Dataset size
> * SFT: 3,150+ training examples (1:1 CN–EN)  and 350 evaluation examples;
> * RLAIF: 9,000+ training examples and 500 evaluation examples (1:1 CN–EN).
>
> ---
>
> **Q3:** clarification about the Qwen2.5-72B model
>
> **A3:** the 72B experiments were still running during submission time. As expected, Qwen2.5-72B performs better than Qwen2.5-32B on the same data. We will add the full 72B results in the revised tables.
>
> ---
>
> **Q4:** additional rules for rejective sampling
>
> **A4:** yes. The rules shown in Figure 1 represent only a subset of our rejective-sampling criteria. Our complete sample-reject pipeline includes the following steps:
>
> * triple trajectory sampling for each query
> * rule-based rejection, including:
>
>   * include all tools and tags
>   * length ≤ 64k tokens
>   * ≥10 reasoning steps and ≥5 tool-use actions
>   * correct tag closure
>   * additional checks such as format validity and language–query consistency
> * Qwen3-based LLM-as-judge filtering to further assess quality
> * topic-stratified human spot-checking, where failed samples are regenerated and re-evaluated
>
> We will include this complete rejective-sampling pipeline in the revised appendix for clarity and reproducibility.
>
> ---
>
> **Q5:** why can this method be considered specifically designed for deep research?
>
> **A5:** unlike daily dialogue or simple QA tasks—where divide-and-conquer provides minimal benefit—deep research requires multi-perspective analysis, evidence gathering, and cross-source verification for a single complex query. This structure naturally aligns with divide-and-conquer. In our framework, each subtask is handled by an agent with multi-tool capabilities, enabling parallel execution and higher-quality synthesis. Compared with general agent workflows, this leads to more comprehensive and reliable final reports. We will clarify this motivation more explicitly in the revised version.
>
> ---
>
> **Q6:** how are w1, w2, R_base, and R_tool calculated?
>
> **A6:** the scoring scheme is summarized below.
>
> *reward weights*
>
> * w1 = 0.9, w2 = 0.1
>
> *base quality reward (R_base)*
>
> * R_base is obtained by having an LLM evaluate the (question, generated report) pair along four dimensions: comprehensiveness, insight, instruction-following, and readability
> * each dimension is scored via a weighted sum of its criteria, and R_base is the average of the four dimension scores
>
> *tool-usage reward (R_tool)*
>
> * let N_calls = min(web_search, crawl_page), with N_min = 2 and N_max = 8
> * R_tool = 0 if N_calls < N_min
> * R_tool = -1 if N_calls > N_max
> * otherwise, R_tool = (N_calls − N_min) / (N_max − N_min)
>
> This design rewards appropriate tool usage and penalizes overuse.
>
> ---
>
> **Q7 and Q8:** missing evaluations and baselines
>
> **A:** the experiments regarding (1) the combined SFT+RL model, (2) baselines from OpenAI DR and Perplexity, and (3) more meaningful comparisons beyond Table 3 are still being completed. We are currently organizing and running the additional evaluations, and we will provide the full results as soon as they are ready. All missing tables and analyses will be added to the final revised version to ensure a complete and comprehensive experimental section.
>
> ---

---

> > ### Comment · Reviewer_o3yX · 2025-11-27
> >
> > Thanks for the response. I believe the next version of the paper will be in better shape by incorporating the above improvements. I will keep my score.

---

### Meta-Review · Area_Chair_bFsE · 2025-12-16

**Summary:**

The submission proposes O-Researcher, a multi-agent synthetic data pipeline and a two-stage training strategy (SFT + agentic RL) for deep research tasks. Reviewers agree the problem is important and the engineering is coherent, but three reviewers (scores 2, 2, 4) raise substantial, unresolved issues: (1) novelty is overstated—there is no new RL algorithm beyond GRPO/PPO-style training with task-specific reward and filtering; the title and claims around \"multi-agent distillation\" mischaracterize what is effectively multi-agent-aided trajectory fine-tuning; (2) the method description remains incomplete and inconsistent, including insufficient details of agent roles/aggregation, missing prompts, and RL specifics (e.g., reward formula lists w3 but only sets w1 and w2; no ablations quantifying the contribution of R_format, R_tool, or GRPO/RLOO choices); (3) heavy reliance on LLM-as-a-judge for rewards risks bias and optimization to style over factuality, with no external validation or human preference baselines; (4) evaluation and generalizability are limited—core results focus on Qwen2.5-32B/72B, with promises (not delivered) to test Llama 3/3.1 and Qwen3; ablations and comparisons against close baselines are missing or superficial; and (5) reproducibility gaps (training budgets, initialization, full hyperparameters, evaluation protocol, prompts) make the work difficult to replicate. The rebuttal clarifies the RL is GRPO with RLOO, adds high-level training configs and a quality-control funnel, admits the lack of algorithmic novelty and revises terminology, and promises broader experiments and complete documentation. However, most fixes are promised rather than demonstrated; core claims remain overstated and critical empirical evidence is absent. I read the author response; the rebuttal did not overcome the objections. I also consulted the provided manuscript text (e.g., Section 3.2, Tables 1–4), which shows expanded baselines but still leaves generalization to other model families, reward component ablations, and reproducibility details insufficient or inconsistent.

**Reviewer Concerns:**

#### Reviewer_XSMC
1. **Concern**: Overstated novelty of the RL component; appears to be standard PPO/GRPO with task-specific rewards.
   - **Why Unresolved**: Authors acknowledged no new RL algorithm and clarified GRPO with RLOO, trajectory filtering, and behavior-corrective rewards. The manuscript still claims a "novel reinforcement learning method" (Abstract, Contributions), and there are no ablations isolating each adaptation’s effect.
   - **Impact on Decision**: Moderate. The paper’s contribution is primarily engineering; novelty claims should be toned down and empirically supported.

2. **Concern**: Generalizability beyond Qwen2.5-32B (e.g., Llama 3/3.1, Qwen3) not demonstrated.
   - **Why Unresolved**: Response promises cross-model experiments; the manuscript includes Qwen2.5-72B and DeepResearchGym results but not Llama/Qwen3 and lacks transfer analysis.
   - **Impact on Decision**: High. Without evidence of transfer to other open-source families, claims of a general pathway for advancing open-source LLMs are not substantiated.

---

#### Reviewer_BMec
1. **Concern**: Algorithmic novelty is limited; RL is essentially PPO-style with LLM-based rewards.
   - **Why Unresolved**: Authors admit the RL is an adaptation of GRPO; manuscript language still overclaims novelty and lacks comparative RL baselines or ablations.
   - **Impact on Decision**: Moderate. Engineering value is present, but algorithmic novelty is insufficient for this venue.

2. **Concern**: Missing implementation details and under-specified agent roles/collaboration; difficult to reproduce.
   - **Why Unresolved**: The rebuttal provides partial configs (e.g., SFT/RL dataset sizes, GPU counts) but still lacks full prompts, aggregation procedures, and evaluation protocol; Section 3.1 aggregation remains under-detailed.
   - **Impact on Decision**: High. Reproducibility is a core criterion; incomplete documentation prevents proper assessment.

3. **Concern**: No quantitative/qualitative assessment of LLM-generated trajectory quality (human verification, bias, diversity).
   - **Why Unresolved**: A funnel (rule checks, LLM judge, spot-checks) is described, but there are no statistics on reject rates, inter-rater reliability, or bias analyses.
   - **Impact on Decision**: Medium. Data quality underpins the approach; lacking evidence weakens the empirical claims.

4. **Concern**: Evaluation limited to Deep-Research-Bench; missing diverse tasks and fair comparison to existing multi-agent/synthetic frameworks.
   - **Why Unresolved**: Table 1 adds several baselines, but many comparisons are not apples-to-apples, and broader tasks and framework parity (e.g., DeepResearcher, AutoGLM-Research) remain promised.
   - **Impact on Decision**: High. Breadth and fairness of evaluation are not yet adequate.

5. **Concern**: Presentation quality is poor; typographical errors; missing references; disproportionate case study section.
   - **Why Unresolved**: Authors promise cleanup; current manuscript still shows inconsistencies (e.g., reward weights w3 undefined in text, figures/tables partially explained).
   - **Impact on Decision**: Medium. The clarity issues impede verification.

---

#### Reviewer_SQtd
1. **Concern**: Misleading "multi-agent distillation" claim; the method is trajectory fine-tuning, not architectural distillation (e.g., MAGDi).
   - **Why Unresolved**: Authors agree and will adjust terminology; the title still says "distillation," and no contrastive/structural distillation is implemented.
   - **Impact on Decision**: Moderate. Mislabeling of methodology contributes to overstated claims.

2. **Concern**: LLM-as-a-judge as primary reward risks bias and optimization to style rather than factual accuracy.
   - **Why Unresolved**: Tool-use reward is added, but no human preference/RM baselines or external verification are provided. No analysis of judge bias or calibration.
   - **Impact on Decision**: Medium. Reward reliability and alignment quality remain uncertain.

3. **Concern**: Undefined "intrinsic difficulty" quantification used for tool-use scoring.
   - **Why Unresolved**: Authors retract this claim; they use fixed Nmin/Nmax rules. The correction is noted, but the original description indicates shifting specifications.
   - **Impact on Decision**: Low. The fix is simple; the broader reward design still lacks validation.

---

#### Reviewer_o3yX
1. **Concern**: Incomplete manuscript; missing baselines, prompts, method details; unclear aggregation; weak evaluation and no ablations.
   - **Why Unresolved**: Some baselines are added (Table 1), but prompts, aggregation specifics, and ablation studies (e.g., reward components, GRPO vs PPO, tool reward sensitivity) remain absent. Case study narrative still minimal; Appendix partially empty.
   - **Impact on Decision**: High. The work is not in a publishable state without substantial additions.

2. **Concern**: Reward function not novel; no ablations; limited aspects considered.
   - **Why Unresolved**: Manuscript introduces R_format and R_tool but provides no ablation or sensitivity analysis. Weighting inconsistency (w1, w2 stated; w3 omitted) persists.
   - **Impact on Decision**: High. Without ablations, the claimed gains cannot be attributed to specific design choices.

3. **Concern**: Unclear pre-training baselines for Qwen2.5 before SFT/RLAIF.
   - **Why Unresolved**: Not reported. No deltas from base models, making attribution of improvements opaque.
   - **Impact on Decision**: Medium. Readers cannot assess the incremental value of each training stage.

**Reviewer Scores:**

Note: The score increase is subjective and personal.

#### Reviewer_XSMC
- **Original Score**: 6
- **Expected Score After Discussion**: 6
- **Rationale**: Clarifications around GRPO/RLOO and the rejective sampling pipeline are useful, but generalization across model families and ablation evidence remain missing. Novelty claims and reliance on LLM-as-a-judge are insufficiently addressed.

---

#### Reviewer_BMec
- **Original Score**: 2
- **Expected Score After Discussion**: 2
- **Rationale**: Authors promise fixes but do not deliver the necessary empirical evidence, ablations, or full reproducibility. The core critique stands: strong engineering, limited novelty, incomplete evaluation.

---

#### Reviewer_SQtd
- **Original Score**: 4
- **Expected Score After Discussion**: 4
- **Rationale**: The rebuttal corrected mislabeling and clarified the tool-use reward, but did not mitigate judge bias concerns or provide stronger methodological insights beyond engineering. Reviewer explicitly kept their score.

---

#### Reviewer_o3yX
- **Original Score**: 2
- **Expected Score After Discussion**: 2
- **Rationale**: Despite added baselines, the manuscript remains incomplete, under-detailed, and lacks ablations and reproducibility artifacts. Reviewer explicitly kept their score.

---

### Decision · Program_Chairs · 2026-01-26

Reject